# Alteration in Melanin Content in Retinal Pigment Epithelial Cells upon Hydroquinone Exposure

**DOI:** 10.3390/ijms242316801

**Published:** 2023-11-27

**Authors:** Takeyuki Nishiyama, Hiroki Tsujinaka, Tetsuo Ueda, Nahoko Ogata

**Affiliations:** Department of Ophthalmology, Nara Medical University, Kashihara 634-8521, Japan

**Keywords:** neovascular age-related macular degeneration, melanin production, hydroquinone, vascular endothelial growth factor, blue light

## Abstract

Abnormal pigmentation or depigmentation of the retinal pigment epithelium (RPE) is a precursor to neovascular age-related macular degeneration (nAMD). In this study, we evaluated the effects of hydroquinone (HQ), the most potent reductant in cigarette smoke, on the melanin production in RPE cells. Induced pluripotent stem cell (iPS)-derived RPE and adult retinal pigment epithelial (ARPE-19) cells were cultured with HQ. Real-time reverse transcription polymerase chain reaction revealed that the expression of melanin-related genes decreased due to the addition of HQ for 1 day. Enzyme-linked immunosorbent immunoassay showed that the concentration of melanin significantly decreased due to the addition of HQ for 24 h. A suspension of RPE cells with HQ for 24 h was prepared, and the absorbance was measured. The absorbance decreased particularly under blue light, suggesting that blue light may reach the choroid and cause choroidal inflammation. Additionally, melanin levels significantly decreased due to the addition of HQ for 1 week. After blue light irradiation on the RPE with HQ for 1 week, the vascular endothelial growth factor in the medium was significantly higher in the HQ group than in the control group. HQ-induced changes in melanin production may be responsible for the uneven pigmentation of the RPE, and these changes may cause nAMD.

## 1. Introduction

Neovascular age-related macular degeneration (nAMD) is a common macular disease that affects older adults and causes vision impairment, leading to blindness [1,2,3,4]. nAMD is characterized by drusen in the macula, accompanied by choroidal neovascularization [4]. In the pathogenesis of nAMD, alterations in retinal pigment epithelium (RPE) function emerge as a pivotal factor. RPE cells contain three types of pigmented granules: melanosomes, lipofuscin, and melanolipofuscin [5]. Among these, the melanin found within melanosomes plays an important role in photoprotection in the RPE cells [6]. Abnormal pigmentation, depigmentation, and hyperpigmentation of the retinal pigment epithelium (RPE) are precursor lesions of nAMD [7]. However, the association between RPE pigment abnormalities in RPE and AMD pathogenesis remains unclear. To understand the pathology of nAMD, it is essential to consider the associated risk factors. Cigarette smoking is one of the major risk factors for AMD [8,9,10]. Cigarette smoke contains over 4000 toxic substances, among which benzene-1,4-diol (hydroquinone [HQ]) is considered one of the most potent reductants and is toxic to humans [9,11]. Cigarette smoking leads to a notable increase in exposure to benzene and its associated compounds [12]. HQ induces oxidative stress in RPE cells [13,14,15] and is closely associated with AMD. Besides oxidative stress, HQ has demonstrated various effects. These include the induction of autophagy defects in the RPE cells [16], nonapoptotic cell death [17], inhibition of nuclear factor kappa B (NF-κB) activity, and suppression of interleukin (IL)-6 and IL-8 cytokine release from the RPE cells [9]. In addition, HQ is widely prescribed by dermatologists as a stain remover because of its strong bleaching properties. HQ inhibits the activity of tyrosinase involved in melanin synthesis [18] and exerts its effect on melanogenesis via the degradation of melanosomes and destruction of melanocytes [19,20,21]. Therefore, HQ could participate to RPE cell pigmentation defects and predisposes these cells to stress. In this study, we aimed to elucidate the effect of HQ on RPE cells using an induced pluripotent stem cell (iPSC)-derived retinal pigment epithelium (iPS-RPE), especially with respect to melanin production.

## 2. Results

### 2.1. HQ Below 2 µM Does Not Affect the Cell Viability

To evaluate the effects of HQ on the viability of RPE cells, the iPS-RPE cells (0.5–2.0 × 10^4^ cells/100 µL in a 96-well plate) were incubated in different concentrations (0–20 µM) of HQ for 24 h. The cell viability was assessed on the RPE and iPS-RPE cells after 24 h incubation with HQ at different concentrations (0–20 μM). While the number of living cells was decreased significantly by exposure to 20 µM of HQ, HQ stimulation at concentrations below 2 μM had no significant effect on the number of RPE living cells (Figure 1). Based on the results of the WST-8 assay, we conducted the following experiments using HQ concentrations below 2 µM in RPE cell culture.

### 2.2. HQ Exposure for 24 h Reduced Expression of Melanin-Production-Related Genes and Melanin Content, Especially in iPS-RPE Cells

Both adult RPE (ARPE-19) and iPS-RPE cells (0.5–2.0 × 10^4^ cells/100 µL in a 96-well plate) were incubated with different concentrations (0–2 µM) of HQ for 24 h. After treatment, the mRNA levels of the melanin-production-related genes, including human tyrosinase (*TYR*), *TYRP1*, and *DCT* [22], were measured using real-time reverse transcription polymerase chain reaction (RT-PCR)*. TYR*, *TYRP1*, and *DCT* were expressed at significantly higher levels in the iPS-RPE cells than in ARPE-19 cells. Figure 2a–e suggests that it is difficult to investigate the changes in melanin production-related genes in ARPE-19 cells. Thus, iPS-RPE cells were used for further analyses. In the iPS-RPE cells, exposure to 0.2/2 µM HQ resulted in decreased transcription of *DCT* and *TYRP1* (Figure 2a,c), while the transcription of *TYR* did not change significantly (Figure 2d). The enzyme-linked immunosorbent immunoassay (ELISA) results revealed that the melanin production was decreased in the iPS-RPE cells exposed to 0.2/2 µM of HQ for 24 h compared to that in the control group (Figure 3).

### 2.3. HQ Exposure for 24 h Reduces Light Transmission through RPE Cells, Particularly in Blue Light

To investigate whether changes in melanin content alter blue light permeability, the light absorbance of suspensions containing iPS-RPE cells exposed to HQ was measured. The light absorbance at 405 nm and 450 nm was decreased significantly in the 2 µM HQ group compared to that in the control group (Figure 4).

### 2.4. Prolonged HQ Exposure Paradoxically Increased Melanin Content, but the Expression of Melanin-Production-Related Genes Did Not Change

To explore the causes of the increased RPE cell pigmentation in the precursor lesions of AMD [7], we examined the changes in melanin production in RPE cells after exposing them to HQ under different conditions. The iPS-RPE cells (0.5–2.0 × 10^4^ cells/100 µL in a 96-well plate) were incubated with different concentrations (0–2 µM) of HQ for 1 week. The ELISA results revealed that the melanin production of iPS-RPE cells was increased in the cells exposed to 2 µM of HQ for 1 week compared to that of the control group or with 0.2 µM of HQ (Figure 5). Additionally, real-time RT-PCR analysis revealed that the mRNA levels of melanin-production-related genes remained unchanged in cells exposed to HQ for 24 h (Figure 6a–c).

### 2.5. Prolonged HQ Exposure Increased the Expression of Melanocortin 1 Receptor (MC1R) mRNA, a Receptor of Melanin Production Cascade

To investigate the underlying cause of the elevated melanin levels due to HQ stimulation, despite the reduction in melanin-production-related genes, we explored the upstream cascade upstream from tyrosine. The iPS-RPE cells (0.5–2.0 × 10^4^ cells/100 µL in a 96-well plate) were incubated under different concentrations (0–2 µM) of HQ for 1 week. Our focus was on *MC1R*, a receptor involved in the melanin production cascade and a trigger for melanin production [23]. The RT-PCR analysis revealed that *MC1R* expression was decreased after exposure to 0.2 µM HQ for 1 week but increased after exposure to 2 µM HQ for the same duration (Figure 7a,b).

### 2.6. Agouti Signaling Protein (ASIP) Addition Suppressed the HQ-Induced Re-Elevation in Melanin Production

To determine whether changes in *MC1R* receptors affect melanin production, Agouti signaling protein (ASIP), an endogenous antagonist of MC1R that leads to the downregulation and upregulation of eumelanogenesis and pheomelanogenesis, respectively [24,25], was added to the iPS-RPE cells (0.5–2.0 × 10^4^ cells/100 µL in a 96-well plate) cultured with 0–2 µM HQ for 1 week.

The results revealed that the addition of ASIP inhibited the increase in melanin production in the 2 μM HQ group (Figure 8), suggesting that *MC1R* may be related to the re-increase in melanin levels in RPE cells.

### 2.7. Blue Light Irradiation of RPE Cells after 1 Week of HQ Addition Significantly Increases Vascular Endothelial Growth Factor (VEGF)-A Secretion

To investigate the effect of the re-elevated melanin levels in RPE cells, the iPS-RPE cells treated with 2 µM of HQ for 1 week were exposed to blue light for 24 h. After treatment, the VEGF concentration in the culture medium was significantly higher in the blue light exposure group than in the control group. In contrast, the VEGF concentration in the culture medium of iPS-RPE cells without HQ addition demonstrated no significant difference between the groups with and without blue light exposure (Figure 9), suggesting that increased melanin levels may be related to VEGF secretion in iPS-RPE cells.

## 3. Discussion

This is the first study to investigate the mechanism by which HQ induces changes in the melanin levels within RPE cells and the resulting changes observed in vitro. Most RPE cell lines, including ARPE-19 cells, which have been used universally for in vitro RPE experiments, differ from actual RPE cells in various aspects, particularly in their lack of pigments [26]. In the present study, significant differences in the expression of melanin-production-related genes were observed between the ARPE-19 and iPS-derived RPE cells. In other words, the strength of our study lies in its exploration of pigmentation level changes using cultured cells that possess pigmentation, a facet previously absent in cultured cells like ARPE-19. Regarding the mechanism underlying the differential effects of HQ between short-term and long-term administration, we discovered that in the short term, hydroquinone causes RPE cells to reduce their melanin levels by down-regulating the expression of the *DCT* and *TYRP1* genes; however, long-term hydroquinone stimulation appears to increase *MCR1* expression, resulting in increased melanin production. Previous studies have established that *MCR1* acts as a receptor for αMSH, promoting melanin production [23]. The reason for the increase in melanin without the upregulation of *DCT* and *TYRP1* remains unclear. However, Tahseen et al. reported that the stimulation of MC1R induces the maturation from pheomelanin to eumelanin via cAMP upregulation [27]. We hypothesize that these functions of MC1R are responsible for the increased melanin production. The finding that the addition of ASIP [24,25] attenuated the increase in melanin production in the HQ-treated RPE cells supports our hypothesis. Notably, the addition of HQ to other pigment cells increased the melanin production in some cases [28]. Additionally, similar changes were observed in the RPE cells.

From the present study, we inferred that exposure to blue light increases VEGF-A expression in areas of increased melanin levels in RPE cells, whereas in areas with decreased melanin levels, blue light passed into the choroid, potentially causing choroidal inflammation. Exposure of the choroid to blue light has been reported to cause the release of cytokines, including VEGF in vivo [29]. In other words, the varying melanin content in pigment epithelial cells may lead to increased secretion of VEGF and other cytokines from both sites, leading to the development of AMD.

Pigment irregularities in the eye may be due to choroidal circulatory disturbances, which result in irregular blood flow and, thus, uneven HQ action. Studies have indicated the occurrence of choroidal circulatory disturbance in AMD-afflicted eyes [30]. It is possible that melanin production is reduced in areas with poor blood flow because of limited HQ exposure in these areas, whereas areas with mild circulatory disturbance are strongly affected by HQ exposure, resulting in excessive melanin production.

Our study had two limitations. First, we did not observe changes in the choroid after the passage of blue light through the retina. This study focused on the behavior of HQ-treated RPE cells, and the extent of light-induced damage to the choroid in instances of reduced melanin in RPE cells remains unknown. Further investigations, including in vivo studies, focusing on the effect of HQ on the choroid are warranted. Second, the influence of other factors related to smoking cannot be disregarded. Smoking contains more than 4000 toxic substances [9,31], and the effects of these substances on the RPE are thought to be complex. When examining the impact of smoking on the RPE, we need to consider the interaction between toxic substances and smoking.

This study indicated that the prior removal of phenolic compounds, including HQ, from smoking may stabilize the RPE pigmentation. HQ has been identified as a toxic and carcinogenic substance in smoking [9,11,14]. Reducing the number of phenolic compounds originating from smoking can prevent its adverse effects. Moreover, several substances have been explored to reduce the toxicity of HQ in the RPE cells. Bhattarai reported that Resvega alleviated HQ-induced oxidative stress in ARPE-19 cells [14]. Ramírez reported that brimonidine prevented HQ damage to RPE and retinal Müller cells [32]. Additionally, Moustafa reported the protective effects of memantine on HQ-treated RPE and human retinal Müller cells [33]. While these substances might be useful for regulating the pigmentation in the RPE cells, further experiments are required to substantiate their efficacy.

## 4. Materials and Methods

### 4.1. Cell Culture

Two human RPE cell lines, ARPE-19 and iPS-RPE, were separately evaluated. The ARPE-19 cells (ATCC^®^ CRL-2302™), a human RPE cell line [34], were obtained from the American Type Culture Collection (Manassas, VA, USA). These cells were grown in Dulbecco’s Modified Eagle Medium (Life Technologies, Carlsbad, CA, USA) and Ham’s F12 medium (Life Technologies) with L-Glutamine and Phenol Red medium (Fujifilm Wako, Tokyo, Japan) containing 10% (*v*/*v*) fetal calf serum, 100 units/mL penicillin G (Fujifilm Wako, Tokyo, Japan), and 100 µg/mL streptomycin (Fujifilm Wako, Tokyo, Japan) as previously described [34]. The iPS-RPE cells were purchased from Phenocell SAS (Grasse, France) and were grown on a BioCoat Matrigel coated plate (Corning Incorporated, Corning, NY, USA) They were cultured in a 7:3 mixture of Dulbecco’s Modified Eagle Medium/Ham’s F12 medium containing a B-27^®^ serum-free supplement (Life Technologies), 100 units/mL penicillin G (Fujifilm Wako, Tokyo, Japan), and 100 µg/mL streptomycin (Fujifilm Wako, Tokyo, Japan) as previously described [35]. The HQ (Fujifilm Wako, Tokyo, Japan) was purchased commercially and dissolved in the culture medium. For the cell death evaluating experiments, ARPE-19 cells were treated with 0–20 µM HQ (Fujifilm Wako, Tokyo, Japan). For the simulation experiments, the iPS-RPE and ARPE-19 cells were incubated for 24 h or 1 week with 0–2 µM HQ. For the inhibition of MC1R experiments, the iPS-RPE cells were treated with ASIP (Abnova Corporation, Taiwan, China) for 24h after 1 week of incubation with HQ. For all experiments, the same amount of culture medium that was used as a solvent of HQ was added to the control cells when HQ was added to create the same conditions.

### 4.2. Measurement of Viable Cell Numbers Using Tetrazolium Salt Cleavage

The ARPE-19 and iPS-RPE cells (0.5–2.0 × 10^4^ cells/100 µL in a 96-well plate) were incubated with or without HQ for 24 h. After treatment, the number of viable cells was assessed using the Cell Counting Kit-8 (Dojindo Laboratories, Mashikimachi, Japan) according to the manufacturer’s instructions, as described previously [36,37,38]. Briefly, WST-8(2-(2-methoxy-4-nitrophenyl)-3-(4-nitrophenyl)-5-(2,4-disulfophenyl)-2H-tetra-zoliummonosodium salt) solution was added to the cells in 96-well plates, and the cells were incubated at 37 °C for 1 h. The optical density of each well was measured at 450 nm (reference wavelength: 650 nm) using the i-Mark™ microplate reader (Bio-Rad Laboratories, Inc., Hercules, CA, USA).

### 4.3. Measurement of Melanin-Production-Related Genes

After 24 h of incubation with HQ and 1 week of incubation with HQ and/or ASIP, the iPS-RPE cells were harvested. The total RNA from both cell types was extracted using the NucleoSpin RNA kit (Takara Bio, Inc., Kusatsu, Japan) according to the manufacturer’s protocol. The cDNA was synthesized from the total RNA using the PrimeScript™ RT Master Mix (Takara Bio, Inc., Kusatsu, Japan). The PCR primers corresponding to nucleotides 267–400 of *TYR* mRNA (NM_000372.5), 3576–3669 of human *DCT* mRNA (NM_001129889.2), 1404–1538 of *TYRP1* mRNA (NM_000550.2), and 485–641 of human *MC1R* mRNA (NM_002386.3) were synthesized. RT-PCR was performed using the SYBR^®^ Premix Ex Taq TMII (Takara Bio, Inc., Kusatsu, Japan) and LightCycler^®^ (Roche Diagnostics, Mannheim, Germany). As an internal control, 1043–1228 of human β-actin mRNA (NM_001101) (Takara Bio, Inc., Kusatsu, Japan) was used.

### 4.4. Measurement of Melanin Protein in the RPE Cells

To assess the degree of melanin expression in the ARPE-19 and iPS-RPE cells, the cells were seeded in 24-well plates at 2.0 × 10^6^ cells/600 µL per well and were cultured with 0–2 µM of HQ for 24 h or 1 week. The total protein was extracted using the Radioimmunoprecipitation (RIPA) Lysis Buffer System (Santa Cruz Biotechnology, Inc., Dallas, TX, USA) and the amount of melanin was measured using a human melanin Quantikine ELISA kit (Wuhan Huamei Biotech Co., Ltd., Wuhan, China), according to the manufacturer’s instructions.

### 4.5. Measurement of Light Absorbance in the RPE Cell Suspension

To evaluate the light-absorbing ability of the RPE cells, the iPS-RPE cells were cultured in 24-well plates at 2.0 × 10^6^ cells/600 µL per well with 2 µM of HQ for 24 h. After cell culture, the suspensions containing the RPE cells were prepared and placed in a 96-well plate. The light absorbance of these suspensions at 405/450/490/570/630 nm was measured using the i-MarkTM microplate reader.

### 4.6. Blue Light Stimulation for RPE Cells

To investigate the effect of blue light on RPE inflammation, cultured iPS-RPE cells were divided into blue-light-blocking and blue light exposure groups. Each group was seeded in a 24-well plate at 2.0 × 10^6^ cells/600 µL and these cells were treated with 0–2 µM HQ for 1 week. After incubation, each group was exposed to a blue light ramp of a 450 nm wavelength (Handy Blue Color LED Light Source, HL-48; AS ONE Corporation, Osaka, Japan) at 232 lumens for 24 h. The blue light ramp was placed 20 cm vertically from the plate. The light exposure duration that would be expected to produce a significant difference in the VEGF-A expression was determined based on previous studies involving blue light exposure in mice [29]. In the light-blocking group, the plate was covered with LED shading tape that can block over 99% of light (Ainex, Tokyo, Japan) to prevent blue light exposure. The cells were harvested immediately after light exposure. The VEGF expression in the medium was measured using the LBIS human VEGF ELISA kit (Fujifilm Wako, Tokyo, Japan) according to the manufacturer’s protocol.

### 4.7. Data Analysis

The results are presented as mean ± standard error (SE). Statistical significance was determined using the Mann-Whitney test using the GraphPad Prism 9.5 software (GraphPad software, La Jolla, CA, USA). All experiments were performed using 4 wells in each group simultaneously, where “*n*” represents one well, and all “*n*” values were derived from the same plate. The experiments were replicated twice to confirm the results.

## 5. Conclusions

Changes in HQ exposure could alter the amount of melanin production in RPE cells. These findings may contribute to the understanding of the development of AMD.

## Figures and Tables

**Figure 1 ijms-24-16801-f001:**
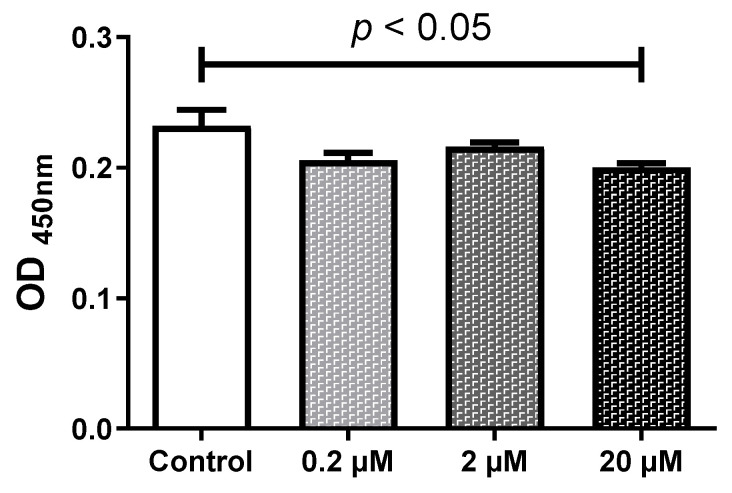
Effect of hydroquinone (HQ) exposure on cultured human adult retinal pigment epithelial (ARPE-19) cells using relative living cells from the control group. The ARPE-19 cells were incubated for 24 h with different HQ concentrations, and the number of living cells was determined using the WST-8 assay. A notable decrease in the number of living cells is observed only with 20 μM HQ compared to that of the control group. Data are presented as means ± standard error of the means for each group (*n* = 4).

**Figure 2 ijms-24-16801-f002:**
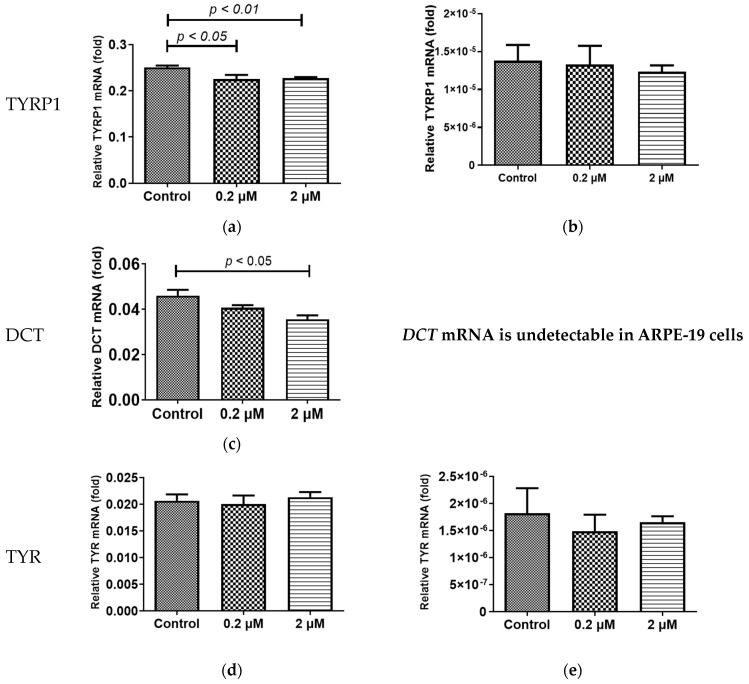
Induction of melanin-production-related genes, including *TYR*, *TYRP1*, and *DCT* expression, due to hydroquinone (HQ) exposure in induced pluripotent stem cell retinal pigment epithelial (iPS-RPE) cells and human adult retinal pigment epithelial (ARPE-19) cells. These cells are exposed to 0–2 μM HQ for 24 h, and the expressions of the mRNAs of *TYR*, *TYRP1*, and *DCT* are determined using real-time reverse transcription polymerase chain reaction, with β-actin used as an endogenous control. The results from iPS-RPE cells are represented in (**a**,**c**,**d**). In the iPS-RPE cells, mRNA levels of *TYRP-1* and *DCT* are significantly decreased in the 2 μM HQ group (**a**,**c**). In contrast, the mRNA levels of *TYR* do not change upon exposure to 2 μM HQ (**d**). ARPE-19 cells exhibit lower expression of *TYR*, *TYRP1*, and *DCT* than that of iPS-RPE cells (**b**,**e**). Notably, *DCT* mRNA is undetectable in the ARPE-19 cells. Data are presented as means ± standard error of the means for each group (*n* = 4).

**Figure 3 ijms-24-16801-f003:**
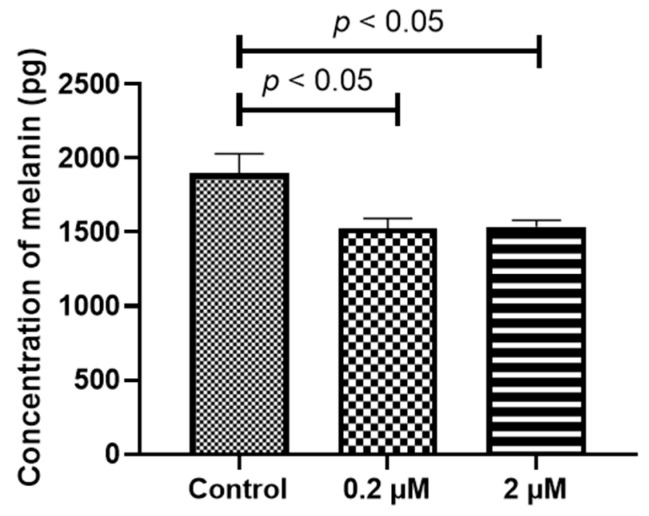
Concentrations of melanin in the induced pluripotent stem cell retinal pigment epithelial (iPS-RPE) cells are determined using enzyme-linked immunosorbent immunoassay. iPS-RPE cells are exposed to 0–2 μM hydroquinone (HQ) for 24 h. Melanin concentration is significantly decreased in the 0.2 and 2 μM HQ groups. Data are presented as means ± standard error of the means for each group (*n* = 4).

**Figure 4 ijms-24-16801-f004:**
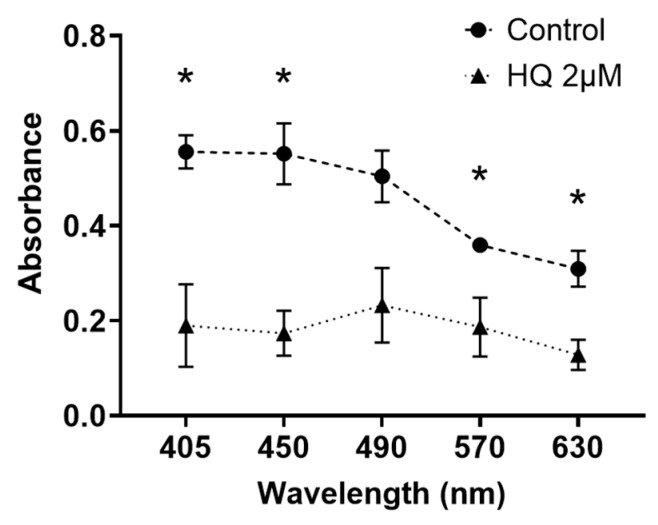
Changes in light transmission in the induced pluripotent stem cell retinal pigment epithelial (iPS-RPE) cells using an absorbance spectrometer. iPS-RPE cells are exposed to 0–2 μM hydroquinone for 24 h and suspensions including RPE cells are prepared. Light absorbance is measured at various wavelengths, including blue light. Data are presented as means ± standard error of the means for each group (*n* = 4). *: *p* < 0.05 vs. Control group.

**Figure 5 ijms-24-16801-f005:**
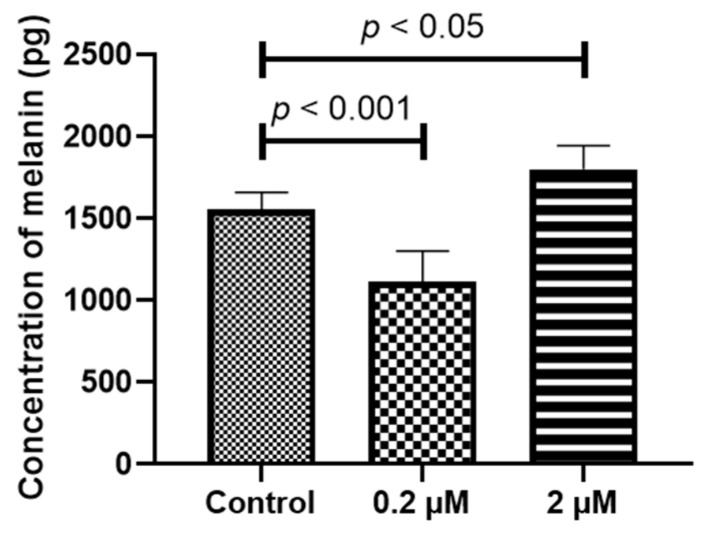
Concentrations of melanin in the induced pluripotent stem cell retinal pigment epithelial (iPS-RPE) cells are determined using enzyme-linked immunosorbent immunoassay. iPS-RPE cells were exposed to 0–2 μM hydroquinone (HQ) for 1 week. The concentration of melanin is significantly decreased in the 0.2 HQ group but increased in the 2 μM HQ group. Data are presented as means ± standard error of the means for each group (*n* = 4).

**Figure 6 ijms-24-16801-f006:**
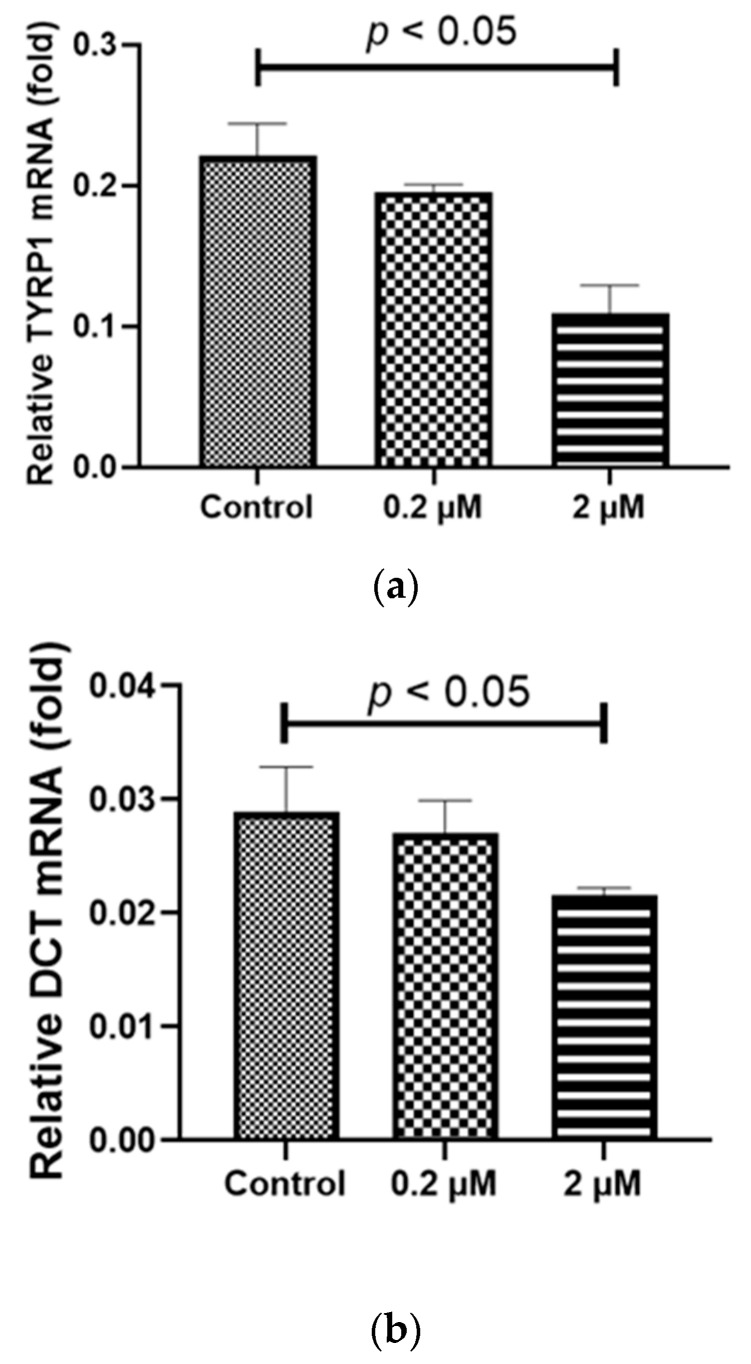
Hydroquinone (HQ)-induced expression of melanin-production-related gene, including *TYR*, *TYRP1*, and *DCT*, expression in induced pluripotent stem cell retinal pigment epithelial (iPS-RPE). These cells were exposed to 0–2 μM HQ for 1 week, and the mRNA expressions of *TYRP1* (**a**), *DCT*, (**b**), and *TYR* (**c**) are determined using real-time reverse transcription polymerase chain reaction using β-actin as an endogenous control. The mRNA levels of *TYRP-1* and *DCT* are significantly decreased in the 2 μM HQ group. In contrast, the mRNA level of *TYR* does not change upon exposure to 2 μM HQ (**c**). Data are presented as means ± standard error of the means for each group (*n* = 4).

**Figure 7 ijms-24-16801-f007:**
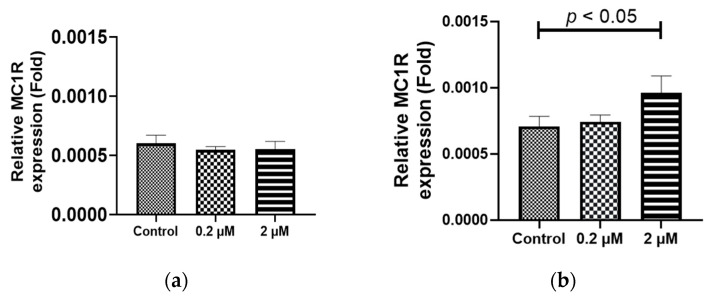
Hydroquinone (HQ) -induced exposure of melanocortin 1 receptor (*MC1R*) in the induced pluripotent stem cell retinal pigment epithelial (iPS-RPE) cells. iPS-RPE cells were exposed to 0–2 μM HQ for 24 h (**a**) or 1 week (**b**). The mRNA level of *MC1R* is significantly increased in the 2 μM HQ group, whereas the mRNA level of *MC1R* does not change in the 2 μM HQ group. Data are presented as means ± standard error of the means for each group (*n* = 4).

**Figure 8 ijms-24-16801-f008:**
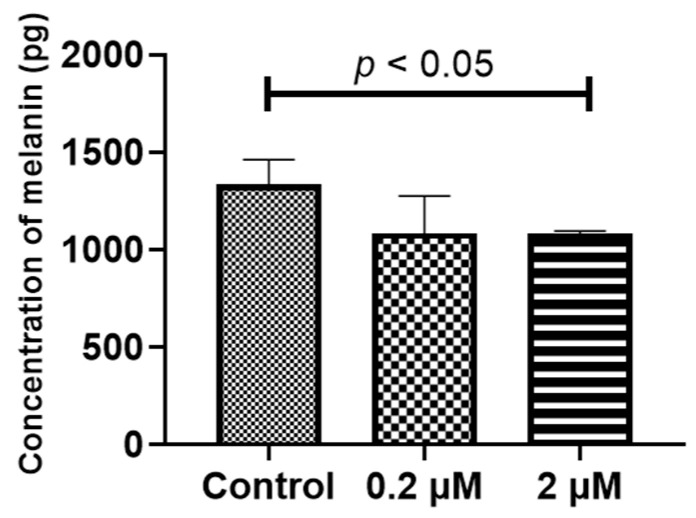
Concentrations of melanin in the induced pluripotent stem cell retinal pigment epithelial (iPS-RPE) cells determined using enzyme-linked immunosorbent immunoassay. iPS-RPE cells are exposed to 0–2 μM hydroquinone (HQ) for 1 week with Agouti signaling protein. The concentration of melanin is significantly decreased in the 2 μM HQ group. Data are presented as means ± standard error of the means for each group (*n* = 4).

**Figure 9 ijms-24-16801-f009:**
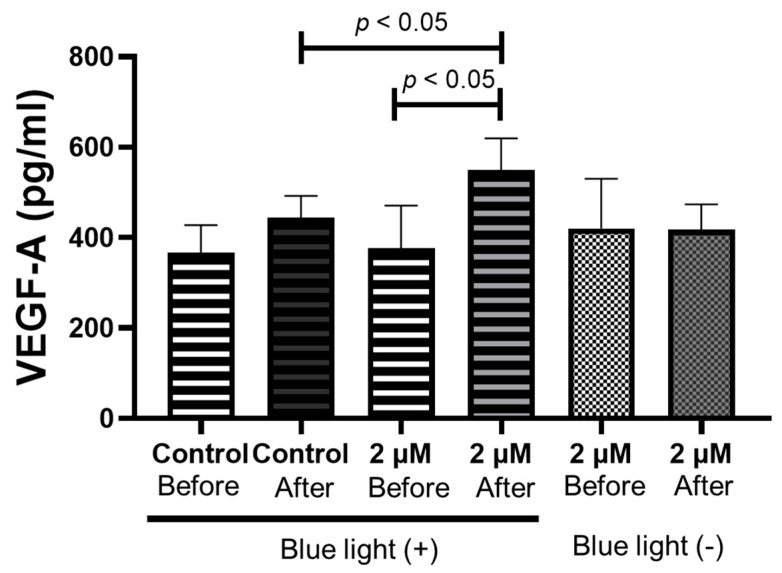
Concentrations of vascular endothelial growth factor (VEGF)-A in the induced pluripotent stem cell retinal pigment epithelial (iPS-RPE) cells determined using enzyme-linked immunosorbent immunoassay. iPS-RPE cells are exposed to 0–2 μM hydroquinone (HQ) for 1 week. After incubation, each group is exposed to blue light ramp for 24 h. The concentration of VEGF-A is significantly increased only in the 2 μM HQ group with blue light exposure. Data are presented as means ± standard error of the means for each group (*n* = 4).

## Data Availability

The data presented in this study are available upon request from the corresponding author.

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
