# Peer review of "Alteration in Melanin Content in Retinal Pigment Epithelial Cells upon Hydroquinone Exposure"

_ijms, 2023, doi:10.3390/ijms242316801_

Round 1

Reviewer 1 Report

Comments and Suggestions for Authors

T

Author Response

Reviewers' comments:

Reviewer #1:

The authors evaluated the effect of hydroquinone (HQ) as a compound found in cigarette

smoke on retinal pigment epithélial cell. HQ is suggestedit to participate to alteration of RPE

cells in AMD’s smokers. The authors investigated the effect of HQ on RPE cells pigmentation

in-vitro. The subject of the study is interesting, however I have several comments.

Introduction:

The introduction is particularly short and not very fluent.

It would be interesting to develop the controversy between the toxic effect of HQ and its use in stain remover. Does it as to do with the dose, the asssociation with other molécules, its mecahnism of action ? is it really « in contrast » ?

Response: As the reviewer pointed out, these two different effects of HQ are not in contrast. Both effects are caused by the same effect: the toxicity of HQ. We have included this point in the revised manuscript (Lines 41-43). And we extended the introduction part and also requested English editing again and to improve the language throughout the revised manuscript.

Is the aim is « to elucidate the mechanism underlying abnormal pigmentation and AMD » or

to elucidate the effetc of HQ on pigmentation.

Response: Thank you for your query. The purpose of this study was to demonstrate the effect of HQ on pigmentation. We have modified this point in the revised manuscript. (Line 44)

It is mentionned you will use iPS-RPE and then you use also ARPE19, which are not pigmented ?

Response: The novelty of our manuscript is that iPS-derived RPE cells, similar cells to the original retinal pigment epithelium, were used to demonstrate differences in pigmentation. To show that ARPE-19, the most commonly used cell-line in basic research of RPE function, is not appropriate for understanding pigmentation, we first used both cell-lines, ARPE-19 and iPS-RPE cells, in the experiment. After showing that only iPS-RPE can demonstrate differences in the pigmentation we used only iPS-RPE cells.

Mat-Meth :

Why did you use ARPE -19 ?

Response: To show that ARPE-19, the most commonly used cell-line in basic research of RPE function, is not appropriate for understanding pigmentation, we first used both cell-lines, ARPE-19 and iPS-RPE cells, in the experiment. After showing that only iPS-RPE can demonstrate differences in pigmentation, we used only iPS -RPE cells.

How do you prepare HQ ?

Response: Culture medium was used as a solvent. We have included an explanation for this process in the revised manuscript (Lines 282-283).

How do you process with your control cells? any treatment at all ?

Response: To make conditions the same in each well, we added the same amount of culture medium that is used as a solvent of HQ to the control cells when we added HQ. We have included this point in the revised manuscript (Lines 288-290).

How do you prepare ASIP? where does it come from?

Response: We purchased ASIP from Abnova Corporation, Taiwan, China. Because ASIP’s original form is liquid, we added ASIP directly to the medium. To make conditions the same, we added the same amount of culture medium to the control cells when we added ASIP. We have included this description in the revised manuscript (Lines 288-290).

What is the interest in measuring melanin in ARPE-19 cells since they are not pigmented ?

Response: As mentioned above, to show that ARPE-19, the most commonly used cell-line in basic research of RPE function, is not appropriate for understanding pigmentation, we first used both cell-lines, ARPE-19 and iPS-RPE cells, in the experiment. After showing that only iPS-RPE can demonstrate differences in pigmentation, we used only iPS -RPE cells.

Blue light stimulation : the light you use is given as light wavelength region: 450 – 500 nm

how is measured the flux (lumens).

It would be more appropriated to give the illuminance received by the cells (meaning luminous

flux received depending on the beam angle and distance).

Response: According to the reviewer’s suggestion, we have described the beam angle and distance from the light to the cells in the revised manuscript (Lines 333-334).

Do you cover the plate during the exposure, can it influence the quality of light reaching the

cells?

Response: In the light-blocking group, the plate was covered with LED shading tape. According to information from the manufacturer, the tape blocks over 99% of light. We have included this information in the revised manuscript (Lines 336-338).

Can you be more precise? In the paper cited

Line 304-305 : you say that you use the amount and duration of exopsure expected to induce

inflammatory response but you are not working on inflammation?

Response: As the reviewer pointed out, we measured only the change in VEGF-A. We have modified this point in the revised manuscript (Line 335).

Line 308 « The cells were harvested after light exposure » : when after exposure ?

Response: The cells were exposed to blue light for 24 h and harvested immediately after exposure. We have modified this point in the revised manuscript (Line 338).

Mainly there is no information on statistics.

Response: Per the reviewer’s observation, we have created a Data analysis section in the revised manuscript (Lines 341-345).

Resultts :

What is "n" exactly? Is it one well or a mean of a triplicate? Were all the "n" on the same plate

or from plates from different experiment at different day?

Response: Per our response to the above concern, we have created a Data analysis section in the revised manuscript (Lines 341-345). Briefly, all experiments were performed with 4 wells in each group simultaneously. “n” means one well and all the “n” was on the same plate. These revisions to the manuscript should make the Results clearer for the reader.

Figure 1: Statistical analysis, what test?

is 0.2 significantly different from 20 μM

Response: Statistical significance was determined by Mann-Whitney test using GraphPad Prism 9.5 software (GraphPad Software, La Jolla, CA). This information is now described in the revised manuscript. (Lines 342-345)

 In Figure 1, the 0.2μM group was not significantly different from the 20μM group.

the data could be expressed as % of survival compared to control, it seems to be about 18%

lower,

Response: According to the reviewer’s suggestion, we modified Figure 1 and used relative proportions of surviving cells in the revised manuscript.

Figure 5 :

how do you explain that control are lower than in figure 3?

0.2μM after 1week is very similar than at 24h incubation.

Response: Because the time from seeding the cells to measuring the amount of melanin differed, it is impossible to directly compare the 24 hours after incubation group with the 1 week after incubation group. In the iPS-cells, pigmentation differed, depending upon the time from seeding the cells to measuring the amount of melanin.

Figure 6 : DCT mRNA but What about the portéin?

You should have Figure 6 A, B and C

Response: We feel that mRNA measurement is sufficient to evaluate gene expression of DCT.

We had Figure6A,B and C in the manuscript.

Figure 7 why 2 graph , you should pool them. If you keep 2 graph you should use the same

scale soi t make it easier to compare.

Response: We would like to retain the two graphs separately in Figure 7, because we demonstrated that MC1R had changed only in the 1 week group, not in the 1 day group. However, per the reviewer’s suggestion, we changed the scale of Figure 7 in the revised manuscript.

Discussion:

Not having the way statistics have been conducted and the meaning of n, the interpretation

of the data cannot be properly believed and so the discussion cannot be evaluated

Response: Per the reviewer’s observation, we have created a Data analysis section in the revised manuscript. (Lines 341-345)

Reviewer 2 Report

Comments and Suggestions for Authors

The manuscript titled "Alteration in Melanin Content in Retinal Pigment Epithelial Cells Upon Hydroquinone Exposure" focuses on the changes in melanin content in induced pluripotent stem cell (iPSC)-derived retinal pigment epithelium (iPs-RPE) and human adult retinal pigment epithelial (ARPE-19) cells after hydroquinone exposure. The authors examined cell viability, melanin production in terms of melanin and melanin production-related genes in both cell lines, changes in light transmission after 24 hours of hydroquinone exposure, the effect of prolonged hydroquinone exposure on melanin production, and the melanocortin 1 receptor (MC1R). The authors also examined the Agouti signaling protein to study the relationship between the increase in MC1R and melanin production. Finally, the authors examined the effect of blue light exposure after one week of hydroquinone exposure on vascular endothelial growth factor (VEGF) secretion.

The manuscript is well written and easy to understand. The experiments performed were sufficient and appropriate for the purpose of the manuscript. Half of the references used in the manuscript are dated.  Regarding the novelty of the manuscript, in my opinion, this is the first time that iPS-RPE and ARPE-19 cells have been exposed to hydroquinone.

In my opinion, the results presented in this manuscript are of interest to a broader community. Despite its great potential, there are some minor problems with the paper, which are addressed below:Abstract: separate units from numbers 1day; 24h.

·         Abstract: Define VEGF.

·         Results: line 63 define ARPE-19

·         Figure 3 caption:  define iPS-RPE.

·         Figure 4 caption:  define iPS-RPE and RPE

·         Figure 5 caption:  define iPS-RPE.

·         Figure 6 caption:  define iPS-RPE.

·         Figure 7 caption:  define iPS-RPE.

·         Figure 8 is missing from the manuscript.

·         Figure 9 caption:  define iPS-RPE.

·         References 2 and 17 do not follow the same pattern as the other references and the journal instructions for authors:

o   Journal Articles:
1. Author 1, A.B.; Author 2, C.D. Title of the article.
Abbreviated Journal Name Year, Volume, page range.

Best regards

Author Response

Reviewer #2:

The manuscript titled "Alteration in Melanin Content in Retinal Pigment Epithelial Cells Upon Hydroquinone Exposure" focuses on the changes in melanin content in induced pluripotent stem cell (iPSC)-derived retinal pigment epithelium (iPs-RPE) and human adult retinal pigment epithelial (ARPE-19) cells after hydroquinone exposure. The authors examined cell viability, melanin production in terms of melanin and melanin production-related genes in both cell lines, changes in light transmission after 24 hours of hydroquinone exposure, the effect of prolonged hydroquinone exposure on melanin production, and the melanocortin 1 receptor (MC1R). The authors also examined the Agouti signaling protein to study the relationship between the increase in MC1R and melanin production. Finally, the authors examined the effect of blue light exposure after one week of hydroquinone exposure on vascular endothelial growth factor (VEGF) secretion.

The manuscript is well written and easy to understand. The experiments performed were sufficient and appropriate for the purpose of the manuscript. Half of the references used in the manuscript are dated.  Regarding the novelty of the manuscript, in my opinion, this is the first time that iPS-RPE and ARPE-19 cells have been exposed to hydroquinone.

In my opinion, the results presented in this manuscript are of interest to a broader community. Despite its great potential, there are some minor problems with the paper, which are addressed below:

Abstract: separate units from numbers 1day; 24h.

Response: We separated units from the numbers as recommended in the revised abstract.

Abstract: Define VEGF.

Response: We have defined VEGF in the revised manuscript. (Lines 20-21)

Results: line 63 define ARPE-19

Response: We have defined ARPE-19 in the revised manuscript. (Lines 68)

Figure 3 caption:  define iPS-RPE.

Response: We have defined iPS-RPE in the revised manuscript. (Lines 115-116)

Figure 4 caption:  define iPS-RPE and RPE

Response: We have defined iPS-RPE and RPE in the revised manuscript. (Lines 129-130).

Figure 5 caption:  define iPS-RPE.

Response: We have defined iPS-RPE in the revised manuscript. (Lines 148-149)

Figure 6 caption:  define iPS-RPE.

Response: We defined iPS-RPE in the revised manuscript. (Lines 160-161)

Figure 7 caption:  define iPS-RPE.

Response: We defined iPS-RPE in the revised manuscript. (Lines 181-182)

Figure 8 is missing from the manuscript.

Response: We confirmed that Figure 8 is now presented in the manuscript. (Line 195)

Figure 9 caption: define iPS-RPE.

Response: We defined iPS-RPE in the revised manuscript. (Lines 214-215)

References 2 and 17 do not follow the same pattern as the other references and the journal instructions for authors:

o   Journal Articles:

  1. Author 1, A.B.; Author 2, C.D. Title of the article. Abbreviated Journal Name Year, Volume, page range.

Response: Per the reviewer’s instruction, we corrected the formatting of References 2 and 17 in the revised manuscript. (Lines 369-370, 407)

Round 2

Reviewer 1 Report

Comments and Suggestions for Authors

Dear Author, 

Your study is interessting and for the first time evaluate the effect of HQ on pigmentation in RPE cells wich is pertinent considering the use of HQ for skin depigmentation. 

However, I still have several concern that you will find in the attached file. 

My main concern is abour the sample size. You answered that n is 1 well and when n=4 there are all on the same plate. Usually on cell culture we are making triplicate and n correspond to the average. For several n the experiment is conducted on several plate from different cell pool.

In addition I have question on the results and you will have to improve the discussion. 

I hope the comment will help you,

Best regards, 

Author Response

Reviewer‘s comments:

Reviewer #1:

Introduction:

Line 42 : Reference 11 and 12 cannot be associated to « is closely associated with AMD »

and should be placed after « HQ induces oxidative stress in RPE cells » .

Response: According to the reviewer’s suggestion, we have modified this point in the revised manuscript. (Line 43)

References showing the correlation between smoking and AMD could be :

Klein R, Klein BE, Linton KL, DeMets DL. The Beaver Dam Eye Study: The relation of agerelated maculopathy to smoking. Am J Epidemiol. 1993;137:190–200 ; Vingerling JR,

Hofman A, Grobbee DE, de Jong PT. Age-related macular degeneration and smoking. The

Rotterdam Study. Arch Ophthalmol. 1996;114:1193–6.

Response: One reference, the Rotterdam Study, had already been included in the manuscript, but the other was omitted. In response to the reviewer’s suggestion, we have now incorporated both references in the revised manuscript (Lines 393-394).

Besides oxidative stress, HQ has been shown to have other effect that should be mentionned in the introduction such as to induce autophagy defect (Abokyi et al., 2021), to have a nonapototic effect (Sharma et al., 2012 Indian J Ophthalmol), to prevent nuclear factor kappa B (NF-κB) activity and the release of interleukin (IL)-6 and IL-8 cytokines from RPE cells (Bertram, K.M.; Baglole, C.J.; Phipps, R.P.; Libby, R.T. Molecular Regulation of Cigarette Smoke InducedOxidative Stress in Human Retinal Pigment Epithelial Cells: Implications for Age-Related Macular Degeneration. Am. J. Physiol. Cell Physiol. 2009, 297, 1200–1210 ; Bhattarai, N.; Korhonen, E.; Toppila, M.; Koskela, A.; Kaarniranta, K.; Mysore, Y.; Kauppinen, A. Resvega Alleviates Hydroquinone-Induced Oxidative Stress in ARPE-19 Cells. Int. J. Mol. Sci. 2020, 21, 2066, Ramírez, C.; Cáceres-del-Carpio, J.; Chu, J.; Chu, J.; Moustafa, M.T.; Chwa, M.; Limb, G.A.; Kuppermann, B.D.; Kenney, M.C. Brimonidine can Prevent in Vitro Hydroquinone Damage on Retinal Pigment Epithelium Cells and Retinal Müller Cells. J. Ocul. Pharmacol. Ther. 2016, 32, 102–108 ; Moustafa, M.T.; Ramirez, C.; Schneider, K.; Atilano, S.R.; Limb, G.A.; Kuppermann, B.D.; Kenney, M.C. Protective Effects of Memantine on Hydroquinone-Treated Human Retinal Pigment Epithelium Cells and Human Retinal Müller Cells. J. Ocul. Pharmacol. Ther. 2017, 33, 610–619).

Response: According to the reviewer’s suggestion, we have added these references in the introduction and discussion sections (Lines 402-403, 406-413, 443-447)

Line 43-47 : I understand that you are mentionning this point to emphase that HQ

affects also pigmentation and that you hypothetized that this effect on pigmentation

is implicated on the effect of HQ in AMD. However the way you introduce this

point as to be improve. I suggest something like :

In addition, HQ is widely prescribed by dermatologists as a stain remover because of

its strong bleaching properties . HQ inhibits the activity of tyrosinase involved in

melanin synthesis (Palumbo A, d’Ischia M, Misuraca G, Prota G (1991) Mechanism of

inhibition of melanogenesis by hydroquinone. Biochim Biophys Acta 1073:85–90) and exerts its effect on melanogenesis through degradation of melanosomes and destruction of

melanocytes (Frenk E, Treatment of melasma with depigmenting agents. Melasma: New

Approaches to Treatment, pp. 9–15. Martin Dunitz Ltd., London, 1995 ; Dooley TP, Topical

skin depigmentation agents: Current products and discovery of novel inhibitors of

melanogenesis. J Dermatol Treat 8: 275–279, 1997 ; Curto et al., Biochemical Pharmacology,

Vol. 57, pp. 663–672, 1999). Therefore, HQ could participate to RPE cell pigmentation

defect and predisposes these cells to stress.

In this study, we aimed to elucidate the the effect of HQ on RPE cells using

induced pluripotent stem cell (iPSC)-derived retinal pigment epithelium (iPS-RPE),

especially with respect to melanin production.

Response: According to the reviewer’s suggestion, we have modified our manuscript. (Lines 47–54)

Math &Meth :

Line 272 : inhibition of what ?

Response: We attempted the inhibition of MC1R using ASIP. We have modified our manuscript accordingly (Line 290)

How long to you treat with ASIP ?

Response: iPS-RPE cells underwent a 24-hour treatment with ASIP (Abnova Corporation, Taiwan, China) following a 1-week incubation with HQ. We have accordingly modified our manuscript (Line 291).

Can you mention : agouti-signaling protein (ASIP), an endogenous antagonist

of melanocortin receptors, leading to the downregulation of eumelanogenesis and the

upregulation of pheomelanogenesis (ref)

Response: In response to the reviewer’s suggestion, we have included information about ASIP in the revised manuscript (Lines 187-189).

Line 331 : "n" in the all figures means one well and all the "n" was were on the same

plate. This is my main concern, usually you supposed to do triplicate and n is the

average of the triplicate. And you do several plate with triplicate for different n.

Response: Because iPS-derived RPE cells are characterized by a very slow growth rate, it is difficult to triplicate all experiments. However, as we understood that repeating same experiments is important to confirm the results, we have duplicated all experiments and confirmed the same results. We have modified our manuscript accordingly (Line 350).

Results :

Line 51-52 : « To evaluate the effects of HQ on the viability of RPE cells, iPS-RPE cells

(0.5–2.0 × 104 51 cells/100 μL in 96-well plate) were incubated in different

concentrations (0–20 μM) of HQ 52 for 24 h. The number of living cells was determined

using the WST-8 assay » should be replaced by « Cells viability was assessed on RPE

cells and iPS-RPE cells after 24h incubation with HQ at different concentrations (0–20

μM). »

Response: According to the reviewer’s suggestion, we have modified the text in the revised manuscript (Lines 59–60).

RT-PCR evaluates transcription and not expression. So Line 75 : replace « expression »

by « transcription » .

Response: According to the reviewer’s suggestion, we have modified the text in the revised manuscript (Lines 82–83).

Figure 9 : « Blue light + , 2µM Before » could be pooled with « Blue Light -, 2µM

Before » ?

Response: In the Blue light – group, the plate was shielded with LED shading tape, whereas in the Blue light + group, the plate was not covered. We differentiated between these two groups to demonstrate that applying tape before blue light exposure does not affect VEGF expression.

 If I understand cells are treated for 1 week with 0 or 2 µM HQ and then they

are exposed or not to light and the harvest is done at the end of the 24h,so there is 4

groups : Control (0 µM HQ) without light (before on your graph) ; 0 µM with light

(after on your graph) ; 2 µM HQ without light ; 2 µM with light. ??

Response: As the reviewer highlighted, we could have considered four groups: Control (0 µM HQ) without light (as depicted before on your graph); 0 µM with light (as shown after on your graph); 2 µM HQ without light; and 2 µM with light. However, in Figure 9, our aim is to illustrate that VEGF-A overexpression occurs only when both blue light irradiation and HQ addition are combined. Therefore, our focus was on demonstrating that blue light irradiation alone or HQ addition alone did not result in an increase in VEGF-A.

In the blue Light groups is there a significant difference between the control after and

the 2 µM after ? If not then HQ has no affect.

Response: In Figure 9, our intention is to illustrate that VEGF-A overexpression occurs only when both blue light irradiation and HQ addition are combined. As noted by the reviewer, a significant difference exists between the control after and the 2 µM after group. We have made modifications to the figure accordingly.

Statistic : you should do ANOVA ?

Response: As highlighted by the reviewer, our manuscript features numerous figures that compare three or more conditions. However, it is important to note that our analyses primarily involve two-group comparisons between the control group and the target group and not multiple comparisons. This is the rationale behind our use of the Mann–Whitney test instead of ANOVA.

Discussion :

I aggree with the fact that this is the first study to evaluate the effect of HQ on the RPE

pigmentation.

The discussion has to be improved :

Line 214 : I don’t agree with « replicate an environment closely resembling the actual

retinal pigment epithelium » . It is a little too much since isoléted cells environnement

is far from the environnement of cells in-vivo. However, you can say that the

advantage of the iPS-RPE cell from ARPE19 cells is that they concerve pigmentation.

Response: According to the reviewer’s suggestion, we have modified the text in the revised manuscript (Lines 222-225)

« long-term hydroquinone stimulation appears to increase MCR1 transcription but it

decreases DCT, TYRP1 transcription. If you consider that transcription and expression

are going the same way (and I am not quite sure of that ?) how do you explain the

increase en melanin ?

Response: The reason for the increase in melanin without upregulation of DCT and TYRP1 remains unclear. However, Tahseen H. et al. reported in their study (Photochem Photobiol. 2015 Jan-Feb;91(1):188-200.) that stimulation of MC1R induces the maturation from pheomelanin to eumelanin through cAMP upregulation. We hypothesize that these functions of MC1R may be responsible for the observed increase in melanin production. These hypotheses have been incorporated into the revised manuscript (Lines 231-235).

If HQ increase melanin production » does it mean that long term HQ would increase

protection and so it would be beneficial ?

Response: Upon reviewing Figure 9, we surmise that excessive pigmentation in the RPE could potentially result in VEGF-A hyperexpression upon exposure to blue light, potentially contributing to wet age-related macular degeneration (wAMD).

Our perspective is that uneven pigmentation, whether excessive or insufficient, may not be conducive to retinal health. We have elaborated on this viewpoint in Lines 239-245 of the manuscript.

How do you explain that at 24h HQ decreases melanine at 0,2 µM but increases it at 2

µM.

Response: In Figure 3, we illustrated that the addition of HQ led to a decrease in melanin protein levels in either the 0.2 µM or 2 µM groups.

How do you explain that you could not detect DCT in ARPE-19. It does not seems to

be in accordance with the littérature.

Response: Given that all experiments were conducted twice to validate the findings, we have confidence in the accuracy of our experimental results.

How do you explain that the relative amount of TYR at one week are 10 time higher

than the one at 24h ?

Response: Because the time from seeding the cells to measuring the amount of melanin differed, it is impossible to directly compare the 24 hours after incubation group with the 1 week after incubation group. In the iPS-cells, pigmentation differed, depending upon the time from seeding the cells to measuring the amount of melanin.

What is the relevance of the HQ concentration used with the one from cigarette smoke?

Response: As you pointed out, the isolated cell environment significantly differs from the in vivo cellular environment. We acknowledge the challenge of directly comparing in vitro HQ concentrations to those found in cigarette smoke. Moreover, there is a lack of literature measuring HQ concentration specifically in the retina. However, existing reports in the literature indicate that concentrations of 20-200 μM HQ induce oxidative stress in ARPE-19 cells. In our manuscript, we primarily utilized a concentration of 2 μM HQ, which is lower than the concentrations reported in previous studies.

Have you seen this reference: Cigarette smoke-related hydroquinone dysregulates

MCP-1, VEGF and PEDF expression in retinal pigment epithelium in vitro and in vivo.

Pons M, Marin-Castaño ME.PLoS One. 2011 Feb 28;6(2):e16722. doi:

10.1371/journal.pone.0016722.

Elevated levels of benzene-related compounds in the urine of cigarette smokers. Ong CN, Lee BL, Shi CY, Ong HY, Lee HP.Int J Cancer. 1994 Oct 15;59(2):177-80. doi: 10.1002/ijc.2910590206.

Response: These references play a crucial role in elucidating the connections between HQ and cigarette smoking. We have included these references in the revised manuscript (Lines 396-397, 403-404).

Line 238-239 : do you have any reference to support this point ?

Response: It is difficult to measure the concentration of HQ in the choroid point by point. However, recently, it has been reported that pathyvessel is strongly related to the pathogenesis of wAMD.

Line 241-242 : do you mean you did not study choroid ?

Response: Our study specifically concentrates on RPE cells in vitro; therefore, we did not investigate the choroid in our manuscript.

Line 245 : focusing on its effects : who or what it « its »

Response: We believe that additional investigations, including in vivo studies, specifically targeting the impact of HQ on the choroid, are warranted. This modification has been made in the manuscript (Line 256).

Line 250-251 : you are goign to far on the interpretaiton of the data from your study

Response: In response to the reviewer's suggestion, we have made modifications to our manuscript (Line 261-262).

Round 3

Reviewer 1 Report

Comments and Suggestions for Authors

Dear authors, 

Thank you for answers or for duplicating the experiments